# Hepatitis B and C viral coinfection and associated factors among HIV-positive patients attending ART clinics of Afar regional state, northeast Ethiopia

**Yemane Mengsteab Hagos**[1]*, **Gebrehiwet Tesfay Yalew**[1], **Hadush Negash Meles**[1], **Ephrem Tsegay**[2], **Mulu Lemelem**[2], **Araya Gebreyesus Wasihun**[2]

1 Department of Medical Laboratory Sciences, College of Medicine and Health Sciences, Adigrat University, Adigrat, Tigrai, Northern Ethiopia, 2 Department of Medical Microbiology and Immunology, College of Health Sciences, Mekelle University, Mekelle, Tigrai, Northern Ethiopia

* mengsteaby40@gmail.com

## Abstract

### Background

Hepatitis B (HBV) and C virus (HCV) coinfection are the major causes of liver-related morbidity and mortality among people living with Human Immunodeficiency Virus (HIV). The burden of hepatitis among HIV-positive individuals has not been studied in the Afar region. Therefore, this study aimed to determine the prevalence of HBV and HCV coinfection and associated factors among HIV-positive patients in Afar Regional State, northeast Ethiopia.

### Methods

A cross-sectional study was conducted on 477 HIV-positive patients between February 2019 and May 2019. A structured and pretested questionnaire was used to collect socio-demographic data and associated factors. Five milliliters of blood was collected, and Hepatitis B surface antigen (HBsAg) and HCV antibodies were detected using rapid test kits. Positive samples were confirmed using enzyme-linked immunosorbent assay (ELISA). Binary and multivariable logistic regression analyses were performed to identify associated factors. Statistical significance was set at $P < 0.05$.

### Results

Among the 477 study participants, 320/477(67.1%) of them were females and 157(32.9%) males. The overall prevalence of HIV-HBV and HIV-HCV coinfection was 25(5.2%) and 7 (1.5%), respectively. Multi-sexual practice was significantly associated with HIV-HBV coinfection (AOR = 5.3; 95% CI: 1.2–24.4, $P = 0.032$).

### Conclusion

The prevalence of both HIV-HBV and HIV-HCV coinfection was intermediate. Multi-sexual practice was significantly associated with HIV-HBV coinfection. Screening of all HIV-positive

**Data Availability Statement:** All datasets generated and/or analyzed during the current study are summarized in the paper.

**Funding:** The author(s) received no specific funding for this work.

**Competing interests:** The authors have declared that no competing interests exist.

**Abbreviations:** AIDS, Acquired Immunodeficiency Syndrome; ALT, Alanine Transaminase; ART, Anti-Retroviral Therapy; CD, cluster of differentiation; CD4, CD4 + T cells (T lymphocyte bearing CD4 receptor); DNA, Deoxyribose Nucleic Acid; ELISA, Enzyme-Linked Immunosorbent Assay; HBsAg, Hepatitis B Surface Antigen; HBV, Hepatitis B Virus; HCV, Hepatitis C Virus; HIV, Human Immunodeficiency Virus; RNA, ribose Nucleic Acid; TDF, : tenofovir; WHO, World Health Organization; 3TC, lamivudine; AZT, zudovidine; Nuv, nuvirapine; EFV, efavirinaze.

patients for HBV and HCV and health education regarding the transmission modes should be considered.

# Background

Hepatitis B virus (HBV) and hepatitis C virus (HCV) are most common causes of viral hepatitis which is an inflammation of the liver cells leading to chronic liver disease [1]. According to recent estimates, there are around 240 million people chronically infected with hepatitis B virus (HBV) in the world where low and middle-income countries are more affected. Whereas, chronic HCV infection accounts for 71 million people worldwide [2,3].

HBV and HCV are transmitted through contaminated blood or body fluids. Additionally, HBV can be acquired through sexual contact, sharing needles or syringes, and from the mother to the child. HCV is transmitted mainly by sharing needles with an infected individual and by blood transfusion [4].

HBV and HCV coinfection are the major causes of liver-related morbidity and mortality among people living with HIV [5]. Globally, 2.7 and 2.3 million of HIV- positive individuals are chronically infected with HBV and HCV, respectively [6]. However, these prevalence rates vary greatly from one country to another country. In Africa, HBV and HCV prevalence rates among HIV-positive patients range from 4.3–23.7% and 0.8–10.3%, respectively [7–11].

The natural course of viral hepatitis infection and the progression of liver disease to cirrhosis and hepatocellular carcinoma are accelerated among HIV-positive individuals [12]. HIV/HBV coinfected groups have a greater median liver enzyme baseline and greater hepatotoxicity than HIV mono-infected group [13]. They are also more likely to develop liver fibrosis and cirrhosis than HBsAg-negative subjects [14,15]. Furthermore, HIV-infected individuals are negatively affected by HCV with increased rate of hepatic deterioration and a persistently increased serum transaminase [16,17].

HBV or HCV coinfection with HIV further increases the risk of end-stage liver disease (ESLD). In particular, patients who are triply infected with HIV, HCV, and HBV have a 12-fold higher incidence rate of ESLD than HIV mono-infected patients, even in the modern ART era [18]. Furthermore, HIV-positive patients on ART with HBV and HCV coinfection show a lower CD4+ T-cell recovery and are associated with significantly higher HBV DNA viral load levels. Moreover, these patients are more likely to develop a risk of AIDS or death than HIV mono-infected persons [19–22].

Available studies in Ethiopia indicate that HBV and HCV coinfection among HIV-infected individuals estimated to be 5.9–42.8% and 1.3–10.5%, respectively [23–28]. However, to the best of our knowledge, this problem has not been studied in the Afar region. Therefore, data in a setting like our study area, where resources for screening, diagnosis, and treatment are limited, are required for regional and national health policy makers to help design strategies for HBV/HCV and HIV coinfection. This study aimed to determine the prevalence and associated risk factors of hepatitis B surface antigen (HBsAg) and HCV antibodies in HIV-positive patients at selected health facilities in the Afar region.

# Materials and methods

## Study area

The study was conducted in health facilities of Afar regional state, which is located in the Great Rift Valley of northeastern Ethiopia. The region is administratively divided into 5 zones,

which are further divided into 29 districts and 2 administrative towns called Samara and Awash Fentale [29]. Samara, the city of Afar, is located 587 km east of Addis Ababa, the capital city of Ethiopia. This study was conducted at Dubti Referral Hospital, Worer Health center and Awash Health Center. Dubti Referral Hospital is positioned 10 km southeast of Samara. Worer Health center is located 300 km southwest of Samara. Awash Health Center the third selected health facility is approximately 360 km Southwest of Samara.

There were approximately 10,664 HIV positive patients served in ART clinics in the region [30]. During the data collection period, approximately 2,160 ART patients were served in the ART clinics of the selected health facilities. The selection of ART drugs was according to the Ethiopian ART guidelines which consists of three first line regimens; [Tenofovir (TDF) + Lamivudine (3TC) + Nevirapine (NVP)/Efavirenz (EFV); or Zidovudine (AZT) +3TC + NVP/EFV; or Stavudine (d4T) +3TC + NVP/EFV] [31].

## Study design and period

This health facility-based cross-sectional study was conducted at selected health facilities in the Afar region from February 2019 to May 2019.

## Source population and study participants

All HIV-positive patients attending ART clinics of Dubti Referral Hospital, Awash Health Center and Worer Health Center were considered as a source population. The study participants were all HIV positive adults who visited the ART clinics during the study period.

## Sample size determination and sampling techniques

The sample size for the study was calculated using the following formula:

$$x = \frac{Z^2 \alpha/2 P(1-P)}{d^2}$$

Considering 6.1% estimated previous prevalence, [24] 2% precision (d = 0.02), and 95% level of confidence (z = 1.96). However, the source population was so small that finite population correction was applied. Therefore, a total number of 482 study participants, including 10% contingency were enrolled in the study.

As the study was conducted in three different health facilities, the sample size was allocated proportionally.

Hence, using a non-probability sampling technique, 482 consecutive study participants were selected from among HIV-positive patients who attended ART clinics at health facilities based on the inclusion criteria of the study.

## Eligibility criteria

### Inclusion criteria.

✓ Individuals above 18 years of age who visited health facilities for ART services during the study period

✓ HIV infected individuals who have provided written informed consent were included.

### Exclusion criteria.

• Individuals who were critically ill, unable to give blood samples and

- Insufficient blood sample were excluded from the study

## Study variables

### Dependent variables.

- Status of Hepatitis B and C virus infection

### Independent variables.

- Socio-demographic characteristics such as age, sex, marital status, residence, educational level, occupation

- Possible Associated factors: multi-sexual practice, blood transfusion history, abortion, history of surgery, home delivery, female genital mutilation, contact with a person with liver disease, dental procedure, piercing, scarification, and tattooing

## Data collection tools

Socio-demographic data and risk factors were collected using a structured questionnaire adapted from similar literatures [23,24,26–28] for this study by trained diploma nurses working at the ART clinic.

## Specimen collection and processing

Five milliliters of blood was aseptically collected from each study participant by vein-puncture, and serum was separated. The specimens were stored at −20˚C upon delay and a cold box was used for transportation.

## Serological testing

Sera were tested for the presence of HBV and HCV infections by testing for HBsAg and HCV Ab, respectively. The presence of HBsAg was determined using commercially available rapid diagnostic tests (RDTs) (Nantong Egens Biotechnology Co., Ltd., China). Antibodies against HCV infection were detected using RDT (Zhejiang Orient Gene Biotech Co., Ltd., United Kingdom). All positive sera were further tested using HBsAg and HCV Ab enzyme-linked immunosorbent assay (ELISA) (Beijing Wantai Biological Pharmacy Enterprise Co. Ltd.) at Mekelle Blood Bank. All tests were performed according to the manufacturer's instructions.

## Quality control

The questionnaire was piloted two weeks before the actual data collection time on 5% (24 patients) of the calculated sample size at Samara health center. Ambiguous words and concepts were corrected accordingly. Training was provided to the data collectors on how to complete the questionnaire and the sampling process. The questionnaires were checked for completeness, and unrecorded values and unlikely responses were manually cleaned.

Standard laboratory guidelines were followed for specimen collection, handling, transportation, laboratory analysis, and interpretation of the test results. HBsAg and anti-HCV Ab-positive and-negative control samples were used to check the quality of both hepatitis B and C kits before conducting the tests.

### Data analysis and interpretation

SPSS version 20 was used for the data entry and analysis. Descriptive statistics such as frequencies, percentages, means, standard deviations and crosstabs were computed to describe the study population in relation to Sociodemographic and associated factors. Bivariate logistic regression was used to assess the association between independent variables and dependent variable. Variables with p < 0.2 in the bivariate logistic regression were imported to multivariate regression analysis to identify the factors that were statistically significant in relation to the dependent variable. Odds ratios and 95% confidence intervals were calculated to determine the strength of association between HBsAg seropositivity and predictors. Statistical significance was set at less than 0.05. Finally, the data were presented with texts and tables.

### Ethical consideration

Ethical clearance was obtained from Mekelle University College of Health Science Ethical Review Committee (ERC 1212/2019). Before the data collection, the Afar Regional Health Bureau wrote a letter of cooperation to the selected Health facilities. Written consent was obtained from the study participants, and all participant information was kept confidential. The patients' physicians or nurses were notified of the positive results. This study was conducted in accordance with the principles of the Declaration of Helsinki.

## Results

### Socio-demographic characteristics

A total of 482 study participants were enrolled and of this 5 were excluded because of insufficient blood samples. About 157 (32.9%) were males and 320 (67.1%) females. The median age was 35 years (ranging from 18–75 years, IQR; 22–48). Approximately 234 (49.1%) of the participants were married and most, 414 (86.8%) were urban residents. Of the study participants 199 (41.7%) could not read and write while 188(39.4%) attended elementary school (Table 1).

### Seroprevalence of hepatitis B and C viruses

Overall 33 (6.9%) HIV-infected patients were positive for HBsAg or anti-HCV antibodies with rapid tests. After retesting all the positive sera with ELISA, one sample positive for HBsAg with rapid test was found to be negative, giving an overall prevalence of 32 (6.7%) viral hepatitis (HBV and HCV) coinfection among HIV positive patients. Among the positive samples, 25 (5.2%, 95%CI 3.6–7.3) were HBsAg positive and seven (1.5%, 95%CI 0.6–2.7%) were Anti-HCV positive. None of the study participants were positive for triple (HBV/HCV/HIV) infections.

### Associated factors of HBV coinfection

In the bivariate logistic regression analysis multi-sexual practice, home delivery,, and participants who attended elementary school has shown significant association with HBsAg seropositivity (Table 2). These variables were entered into the multivariate logistic regression model. In the final model multi-sexual practice (AOR = 5.3; 95% CI: 1.2–24.4, *P* = 0.032) was significantly associated with HIV-HBV coinfection (Table 3).

## Discussion

Information on the prevalence of viral hepatitis coinfection among HIV-infected individuals is vital for guiding government policymakers and stakeholders for efficient management

**Table 1. Bivariate analysis of socio-demographic characteristics of HBV infection among HIV positive individuals.**

| Variables | | N = 477(%) | ELISA HBV +ve, n (%) | COR (95% CI) | P-value |
|---|---|---|---|---|---|
| Sex | Female | 320 (67.1) | 16 (5.0) | 1 | |
| | Male | 157 (32.9) | 9 (5.7) | 1.2(0.499–2.68) | 0.736 |
| Age | 18–29 | 117 (24.5) | 6 (5.1) | 1.6(0.31–8.01) | 0.589 |
| | 30–39 | 186 (39.0) 114 (23.9) | 11 (5.9) | 1.8(0.39–8.45) | 0.443 |
| | 40–49 | 60 (12.6) | 6 (5.3) | 1.6(0.32–8.24) | 0.567 |
| | ≥50 | | 2 (3.3) | 1 | |
| Marital status | Single | 90 (18.9) | 5 (5.6) | 1.5(0.28–7.86) | 0.652 |
| | Married | 234 (49.1) | 15 (6.4) | 1.7(0.379–7.7) | 0.484 |
| | Divorced | 101 (21.2) | 3 (3.0) | 0.77(0.12–4.73) | 0.773 |
| | Widowed | 52 (10.9) | 2 (3.8) | 1 | |
| Residence | Rural | 63 (13.2) | 5 (7.9) | 1.7(0.61–4.7) | 0.308 |
| | Urban | 414 (86.8) | 20 (4.8) | 1 | |
| Educational level | Illiterate | 199 (41.7) | 15 (7.5) | 1 | |
| | Elementary | 188 (39.4) | 6 (3.2) | 0.40(0.15–1.07) | 0.067* |
| | Secondary | 90 (13.4) | 4 (4.4) | 0.57(0.18–1.77) | 0.331 |
| Occupation | Civil servant | 92 (19.3) | 2 (2.2) | 1 | |
| | Merchant | 81 (17) | 4 (4.9) | 2.34(0.42–13.11) | 0.334 |
| | Others[a] | 142 (29.8) | 12 (8.5) | 4.15(0.91–19.0) | 0.366 |
| | No work | 162 (34) | 7(4.3) | 2.0(0.41–9.99) | 0.383 |

[a]Farmer, commercial sex worker, driver, daily worker and student.

* Statistically significant at p≤ 0.2.

**Table 2. Bivariate analysis of risk factors of HBV infection among HIV positive individuals.**

| Variables | | N = 477(%) | ELISA HBV, N = 25(%) | COR (95% CI) | P-value |
|---|---|---|---|---|---|
| Multi-sexual practice | No | 175 (36.7) | 2 (1.1) | 1 | |
| | Yes | 302 (63.3) | 23(7.6) | 7.1(1.626–30.68) | 0.009* |
| Blood transfusion | No | 448 (93.9) | 23(5.1) | 1 | |
| | Yes | 29 (6.1) | 2(6.9) | 1.9(0.37–9.74) | 0.443 |
| Abortion | No | 228 (73.1) | 12 (5.3) | 1 | |
| | Yes | 84 (26.9) | 4 (4.8) | 0.9(0.28–2.89) | 0.859 |
| Hospital Admission | No | 326 (68.3) | 18(5.5) | 1 | |
| | Yes | 151 (31.7) | 7 (4.6) | 0.63(0.23–1.75) | 0.377 |
| History of surgery | No | 428 (89.5) | 22 (4.7) | 1 | |
| | Yes | 49 (6.1) | 3 (6.1) | 1.3(0.306–5.1) | 0.755 |
| Home delivery | No | 176 (63.5) | 12 (6.8) | 1 | |
| | Yes | 101 (36.5) | 3 (3) | 0.42(0.12–1.52) | 0.185* |
| Dental procedure | No | 350 (73.4) | 16(4.6) | 1 | |
| | Yes | 127 (26.6) | 9 (7.1) | 1.44(0.60–3.47) | 0.417 |
| Female genital mutilation | No | 92 (28.6) | 3(3.3) | 1 | |
| | Yes | 230 (71.4) | 12 (5.2) | 1.6(0.45–5.93) | 0.456 |
| Tattooing | No | 319 (66.9) | 15 (4.7) | 1 | |
| | Yes | 158 (33.1) | 10 (6.3) | 1.1(0.45–2.7) | 0.837 |
| Piercing | No | 164 (34.4) | 10 (6.1) | 1 | |
| | Yes | 313 (65.6 | 15 (4.8) | 0.75(0.31–1.8) | 0.515 |
| Scarification | No | 336 (70.4) | 14 (4.2) | 1 | |
| | Yes | 141 (29.6) | 11 (7.8) | 1.7(0.72–3.9) | 0.206 |
| Contact with a person having liver disease | No | 421 (88.3) | 21 (5.0) | 1 | |
| | Yes | 56 (11.7) | 4 (7.1) | 1.5(0.46–4.66) | 0.517 |

COR crude odds ratio.

* Statistically significant at $P < 0.2$.

Table 3. Multivariate analysis of the variables associated with HBV infection.

| Variables, | N = 477 | AOR (CI 95%) | P-value |
|---|---|---|---|
| **Multi-sexual practice** | No | 1 | |
| | Yes | 5.3(1.2–24.4) | 0.032 |
| **Home delivery** | No | 1 | |
| | Yes | 0.3(0.08–1.1) | 0.063 |
| **Educational status** | No | 1 | |
| | Yes | 0.6(0.263–1.499) | 0.294 |

AOR Adjusted odds ratio.

*statistically significant at P < 0.05.

strategies, prevention, treatment, and controlling mechanisms. However, in resource-limited regions, such as our study areas, the magnitude and severity of coinfection with HBV and HCV among HIV-infected patients is not well documented.

The HBsAg prevalence in HIV positive patients in our study area (5.2%), is in line with other studies in other regions in Ethiopia: Southern Ethiopia (6.3%) [23], Amhara region (6.1%) [24] and (5.6%) [26], Tigrai region (5.9%) [27], and abroad in Korea (5.0%) [32]. This is also similar to the national prevalence of HBsAg among the HIV positive participants reported in a systematic review of hepatitis viruses in Ethiopia (5.2%) [33].

In contrast, it is lower than studies conducted at Bale Zone, Southeast Ethiopia, among out patients of Goba General +Hospital (42.3%) [28], Kenya (10.3%) [8], Cameroon (23.7%) [11], Gambia (12.2%) [34], Cambodia (11%) [20] and Gabon (8.8%) [21].This discrepancy could probably be due to the difference in the study participants and specific risk factors [28], the relatively large sample size used [20] and the more sensitive diagnostic methods they used, ELISA and molecular techniques [8,11]. Moreover, the study participants were ART naïve [11], since patients on TDF based ART have higher seroconversion rate of HBsAg than those ART naïve patients [22,35].

Our results were however, higher than reports from Kenya (4.26%) [36], Tanzania (2.3%) [37], Cameroon (1.16%) [38], Libya (0.03%) [39], Italy (4.1%) [40] and Brazil (3.1%) [22]. This might be due to the differences in socio-demographic characteristics, in hepatitis epidemiology of the countries: the lower endemicity [22,40], methodological differences [37], the community-based nature of the studies [38,39].

In this study, the prevalence of HIV-HCV coinfection was 1.5%, which is comparable to other studies conducted in Amhara region, northwest Ethiopia (1.3%) [24] and Korea (1.7%) [32]. In contrast to this study, higher seroprevalence was reported in other regions of the Ethiopia, Tigrai (6.6%) [41], Amhara (5.0%) [26] and Southern Ethiopia (3.2%) [23], abroad in Kenya (6%) [8], Cameroon (7.2%) [11], Cambodia (5.3%) [20], Ukraine (12.1%) [43], Italy (7.8%) [39] and Brazil (9.7%) [42]. However, our finding was higher than those in Libya (0.15%) [39] and Kenya (0.46%) [36]. The relatively low prevalence of HCV in this study compared to other studies might be due to the absence of the most common predictor of HCV infection, injection drug use practice (IDU), which is considered relatively uncommon in Ethiopia. IDU is strongly associated with a high prevalence of HCV [40,42,43]. Blood transfusion is another factor considered to be associated with a high prevalence of Hepatitis C, as shown in other studies [39,44]. Studies from Ethiopia [24,41] also showed that blood transfusion was a risk factor for HIV-HCV coinfection.

In the present study, local risk factors for hepatitis co-infection were investigated hence multi-sexual practice was significantly associated with higher HBsAg sero-positivity (AOR = 5.3; 95% CI: 1.2–24.4, P = 0.032). This is in agreement with other studies [11,24,27,28]

which also demonstrated a history of multiple sexual partners to be significantly associated with HBV coinfection among HIV positive patients.

Although, statistical significant association was not shown a high prevalence of HBsAg was observed in the age group of 30–39. This is not in concordance with other studies that revealed that individuals within the 40–49 age categories were the most affected group [23,26]. Furthermore, participants who attended elementary school were less likely to test positive for HBsAg, which is not in agreement with another study in which elementary education level was shown to be an independent risk factor for HIV-HBV coinfection [22].

This study has limitations. Owing to the absence and incomplete data, we could not include and assess some relevant clinical data, such as CD4 count and viral load. Moreover, we could not identify the factors associated with HCV-HIV coinfection because of the small number of HCV-HIV coinfected patients. Finally, HBV DNA and HCV RNA levels were not evaluated due to lack of molecular virology laboratory facilities.

## Conclusions and recommendations

The prevalence of both HIV-HBV and HIV-HCV coinfection in this cross-sectional study was intermediate. Multi-sexual practice was significantly associated with HIV-HBV coinfection. Therefore, for better management of HIV-positive patients screening of all HIV positive patients for hepatitis B and C should be considered. Health education regarding the modes of HBV transmission of the hepatitis B should also be provided at health facilities.

## Supporting information

**S1 File. Annexes including information sheet, informed consent, and questionnaire.**
(DOCX)

**S2 File. A scanned pdf file of Ethical clearance.**
(PDF)

## Acknowledgments

We are grateful to all the staff members at the Afar Regional Laboratory and health facilities for their cooperation during this study. We would like to thank Mrs. Tsega Kahsay for her help in the laboratory during ELISA tests. We are also grateful to the Mekelle Blood Bank for their cooperation with the ELISA kits.

## Author Contributions

**Conceptualization:** Yemane Mengsteab Hagos, Araya Gebreyesus Wasihun.

**Data curation:** Yemane Mengsteab Hagos.

**Formal analysis:** Yemane Mengsteab Hagos, Gebrehiwet Tesfay Yalew, Hadush Negash Meles.

**Investigation:** Yemane Mengsteab Hagos, Gebrehiwet Tesfay Yalew.

**Methodology:** Yemane Mengsteab Hagos, Gebrehiwet Tesfay Yalew, Hadush Negash Meles.

**Supervision:** Ephrem Tsegay, Mulu Lemelem, Araya Gebreyesus Wasihun.

**Writing – original draft:** Yemane Mengsteab Hagos, Hadush Negash Meles.

**Writing – review & editing:** Yemane Mengsteab Hagos, Gebrehiwet Tesfay Yalew, Hadush Negash Meles.

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
