## [Decision Letter · Decision Letter 0]

26 Oct 2023

PONE-D-23-22669Hepatitis B and C viral coinfection and associated factors among HIV positive patients attending ART clinics of Afar regional state, northeast EthiopiaPLOS ONE

Dear Dr. Hagos,

Thank you for submitting your manuscript to PLOS ONE. After careful consideration, we feel that it has merit but does not fully meet PLOS ONE’s publication criteria as it currently stands. Therefore, we invite you to submit a revised version of the manuscript that addresses the points raised during the review process.

We look forward to receiving your revised manuscript.

Kind regards,

Jason T. Blackard, PhD

Academic Editor

PLOS ONE

Additional Editor Comments:

This is a cross-sectional study of hepatitis B and hepatitis C in HIV positive persons in Ethiopia.

The study design is standard.  The manuscript should be revised by a native English speaker and/or a professional editing service.

The patients are presenting to several ART clinics; yet, information on specific ART regimens is never provided.  Since some HIV regimens are active against HBV, this is important information to include.

A number of studies related to HBV and HCV prevalence have been conducted in Ethiopia and cited by the authors.  It is quite unclear why another such study was needed and how the current manuscript is similar to or distinct from those previous studies.

The study limitations should also state that HBV DNA and HCV RNA levels were not evaluated.

Reviewers' comments:

Reviewer's Responses to Questions

**Comments to the Author**

1. Is the manuscript technically sound, and do the data support the conclusions?

Reviewer #1: No

Reviewer #2: Yes

2. Has the statistical analysis been performed appropriately and rigorously? 

Reviewer #1: No

Reviewer #2: Yes

3. Have the authors made all data underlying the findings in their manuscript fully available?

Reviewer #1: Yes

Reviewer #2: Yes

4. Is the manuscript presented in an intelligible fashion and written in standard English?

Reviewer #1: No

Reviewer #2: Yes

5. Review Comments to the Author

Reviewer #1: 1.The paper needs to be revised for English and grammar editing (e.g., line 105-108, 184, 208).

2.Line 94 samara should be written in capital letter (southwest of Samara)

3.Line 96 and 98 clients should be changed to participants or patients.

4.Line 106: It is not clear what is meant by Source population.

5.Line 105-108-The paragraph need to be rephrased.

6.It is not clear how long the participant was on ART and whether their medications included treatment that can treat HBV.

7.Line 127: It is not clear why vaccinated individuals were excluded from the study since they did not check for other serological markers. Kindly clarify in the manuscript whether participants’ vaccination history was self-reported only or confirmed using clinical records. The limitations of relying on self-reported clinical history / patient recall, should be adequately addressed in the manuscript where appropriate.

8.The authors allude to the use of a questionnaire in collecting relevant participant data. For validation purposes, kindly indicate of this questionnaire was piloted. Was a previously published questionnaire used during this study (in which case it should be appropriately cited and referenced) or if a study specific one was developed, can a template be provided as part of supplementary material.

9.The statistical analysis is not in deep described and is not clear which statistical analysis was performed for descriptive analysis.

10.Line 198-199: It is not clear whether the patient who tested negative by ELISA was HCV or HBV positive by rapid. The author should include the rapid results first.

11.Did the authors investigate whether those who have multi-sexual were commercial sex workers or not?

12.Table 1: p value of 0.206 in the variable scarification variables is more than 0.2; Second. & above should read secondary in the educational level variable; Indicate how many were commercial sex worker since it is a high risk for infection under the occupation variable.

13.Write all the abbreviations in full

14.Line 272: It is not clear if Egypt screened blood transfusion for HBV/HCV/HIV. If it is done, when was it introduced? Did the participant indicate when they were transfused? If blood is screened for viruses there might be the reason why HCV is low.

15.Line 276-282: There was no statistical significance with age and educational level.

16.The discussion section should be reorganized. Discussion section should relate to the literature review and research questions and making an argument in support of the overall conclusion.

Reviewer #2: The study entitled “Hepatitis B and C viral co-infection and associated factors among HIV positive patients attending ART clinics of Afar regional state, northeast Ethiopia” was aimed to describe the

prevalence of HBV and HCV coinfection and associated factors among HIV-positive patients in the region of Ethiopia.

The authors are commended on their succinct and foundation of evidence for hepatitis B and C in HIV co-infected patients in a region in Ethiopia.

Introduction:

It is recommended that the introduction should include more on the testing practice for viral hepatitis B and C in the country, some information on the vaccine policy and coverage for hepatitis B and treatment accessibility for hepatitis B and C, mono-infected and co-infected patients in the country. The added information will lead to an understanding to the findings and discussion in relation to age groups, gender and risks.

There were a few grammatical errors in the document and it is suggested that the authors go through and correct.

Page 3, line 44 and 46, change “peoples” to people

Page 6, line 94, use capital “S” fro Samara

Page 7, line 112, is the estimated 6.1% prevalence for HIV? HBV? HCV? Or co-infections?

Page 9, line 137, Is “multi-sexual practice” homosexual and heterosexual practice with multiple partners? Please may you define or provide reference to this?

Page 10, line 148, add “a” to test tube

Page 17, line 243, “O” in “our” should be in small letter “o”.

Page 18, line 252, our results ‘were” however….

6. PLOS authors have the option to publish the peer review history of their article (what does this mean?). If published, this will include your full peer review and any attached files.

Reviewer #1: No

Reviewer #2: No

---

## [Author Response · Author response to Decision Letter 0]

29 Feb 2024

Part One: Point by point response for reviewers

Response for reviewer 1

Comment 1: The paper needs to be revised for English and grammar editing (e.g., line 105-108, 184, 208)

Response 1: We have tried to edit grammar errors by deeply revising the manuscript and by using online editor- grammar checker.

Comment 2: Line 94 samara should be written in capital letter (southwest of Samara)

Response 2: We have written in capital letter accordingly 

Comment 3: Line 96 and 98 clients should be changed to participants or patients.

Response 3: We have replaced “clients” by “patients”

Comment 4: Line 106: It is not clear what is meant by Source population.

Response 4: We have tried to make clear and easily understandable

Comment 5: Line 105-108-The paragraph need to be rephrased.

Response 5: we have rephrased the paragraph 

Comment 6: It is not clear how long the participant was on ART and whether their medications included treatment that can treat HBV.

Response 6: Dear reviewer, we didn’t assess how long the participants were on ART, but we have tried to clarify the medications included that can also treat HBV.

Comment 7: Line 127: It is not clear why vaccinated individuals were excluded from the study since they did not check for other serological markers. Kindly clarify in the manuscript whether participants’ vaccination history was self-reported only or confirmed using clinical records. The limitations of relying on self-reported clinical history / patient recall should be adequately addressed in the manuscript where appropriate.

Response 7: We have modified the exclusion criteria 

Comment 8: The authors allude to the use of a questionnaire in collecting relevant participant data. For validation purposes, kindly indicate of this questionnaire was piloted. Was a previously published questionnaire used during this study (in which case it should be appropriately cited and referenced) or if a study specific one was developed, can a template be provided as part of supplementary material.

Response 8: Modified and corrected

Comment 9: The statistical analysis is not in deep described and is not clear which statistical analysis was performed for descriptive analysis.

Response 9: We have tried to describe briefly the descriptive analysis ( 

Comment 10: Line 198-199: It is not clear whether the patient who tested negative by ELISA was HCV or HBV positive by rapid. The author should include the rapid results first.

Response 10: We have included the rapid test results and the discordant test result is clarified

Comment 11: Did the authors investigate whether those who have multi-sexual were commercial sex workers or not?

Response 11: Dear reviewer, we agreed with your idea. But we only investigate multi-sexual practice.

Comment 12: Table 1: p value of 0.206 in the variable scarification variables is more than 0.2; Second. & above should read secondary in the educational level variable; Indicate how many were commercial sex worker since it is a high risk for infection under the occupation variable.

Response 12: Accepted and Modified

Comment 13: Write all the abbreviations in full

Response 13: We have tried to write all the abbreviations in full

Comment 14: Line 272: It is not clear if Egypt screened blood transfusion for HBV/HCV/HIV. If it is done, when was it introduced? Did the participant indicate when they were transfused? If blood is screened for viruses there might be the reason why HCV is low.

Response 14: Dear reviewer, we agreed with your idea. But we only investigate the history of Blood transfusion

Comment 15: Line 276-282: There was no statistical significance with age and educational level.

Response 15: Comment accepted and modified

Comment 16: The discussion section should be reorganized. Discussion section should relate to the literature review and research questions and making an argument in support of the overall conclusion.

Response 16: We have tried to reorganize and modify it

Response for reviewer 2

Comment 1: It is recommended that the introduction should include more on the testing practice for viral hepatitis B and C in the country, some information on the vaccine policy and coverage for hepatitis B and treatment accessibility for hepatitis B and C, mono-infected and co-infected patients in the country. The added information will lead to an understanding to the findings and discussion in relation to age groups, gender and risks.

Response 1: we have tried to include some information on the testing, vaccine and treatment accessibility in the country 

Comment 2: Page 3, line 44 and 46, change “peoples” to people

Response 2: comment accepted and corrected

Comment 3: Page 6, line 94, use capital “S” for Samara

Response 3: comment accepted and corrected

Comment 4: Page 7, line 112, is the estimated 6.1% prevalence for HIV? HBV? HCV? Or co-infections?

Response 4: comment accepted and modified

Comment 5: Page 9, line 137, Is “multi-sexual practice” homosexual and heterosexual practice with multiple partners? Please may you define or provide reference to this?

Comment 6: Page 10, line 148, add “a” to test tube

Response 6: comment accepted and corrected

Comment 7: Page 17, line 243, “O” in “our” should be in small letter “o”.

Response 7: comment accepted and corrected

Comment 8: Page 18, line 252, our results ‘were” however….

Response 8: comment accepted and corrected.

Part Two: Point by point responses to editor and editorial staffs

Comment 1: Please ensure that your manuscript meets PLOS ONE's style requirements, including those for file naming.

Response 1: comment accepted and we have tried to meets PLOS ONE's style requirements

Comment 2: Consider depositing your raw data in a repository to ensure your work is read, appreciated and cited by the largest possible audience.

Response 2: Comment accepted and raw data will be depositing up on request.

Comment 3: In your Data Availability statement, you have not specified where the minimal data set underlying the results described in your manuscript can be found.

Response 3: supporting information is uploaded 

Comment 4: Your ethics statement should only appear in the Methods section of your manuscript. 

Response 4: ethics statement is written in the methods section.

---

## [Decision Letter · Decision Letter 1]

4 Apr 2024

Hepatitis B and C viral coinfection and associated factors among HIV-positive patients attending ART clinics of Afar regional state, northeast Ethiopia

PONE-D-23-22669R1

Dear Dr. Hagos,

We’re pleased to inform you that your manuscript has been judged scientifically suitable for publication and will be formally accepted for publication once it meets all outstanding technical requirements.

Kind regards,

Jason T. Blackard, PhD

Academic Editor

PLOS ONE

Additional Editor Comments (optional):

None

Reviewers' comments:

Reviewer's Responses to Questions

**Comments to the Author**

1. If the authors have adequately addressed your comments raised in a previous round of review and you feel that this manuscript is now acceptable for publication, you may indicate that here to bypass the “Comments to the Author” section, enter your conflict of interest statement in the “Confidential to Editor” section, and submit your "Accept" recommendation.

Reviewer #1: All comments have been addressed

Reviewer #2: All comments have been addressed

2. Is the manuscript technically sound, and do the data support the conclusions?

Reviewer #1: Yes

Reviewer #2: Yes

3. Has the statistical analysis been performed appropriately and rigorously? 

Reviewer #1: Yes

Reviewer #2: Yes

4. Have the authors made all data underlying the findings in their manuscript fully available?

Reviewer #1: Yes

Reviewer #2: Yes

5. Is the manuscript presented in an intelligible fashion and written in standard English?

Reviewer #1: Yes

Reviewer #2: Yes

6. Review Comments to the Author

Reviewer #1: The author has addressed all the comments satisfactorily and the manuscript should now be accepted for publication.

Reviewer #2: most of the pertinent comments by the reviewers were addressed by the authors. there are several limitations which were outlined by the authors. It is still difficult to determine the extent of risk of transmission as numbers were small, as alluded by the authors. However, the association output was interesting findings for the region.

7. PLOS authors have the option to publish the peer review history of their article (what does this mean?). If published, this will include your full peer review and any attached files.

Reviewer #1: No

Reviewer #2: No

---

## [Editor Report · Acceptance letter]

3 May 2024

PONE-D-23-22669R1 

PLOS ONE

Dear Dr. Hagos, 

I'm pleased to inform you that your manuscript has been deemed suitable for publication in PLOS ONE. Congratulations! Your manuscript is now being handed over to our production team.

Kind regards, 

on behalf of

Dr. Jason T. Blackard 

Academic Editor

PLOS ONE